# Adapting the DIST-M Model for Designing Experimental Activities—A Theoretical Discussion from an Interdisciplinary Perspective

Stefania Lippiello [1,*] and Alessandra Boscolo [2,*]

1   Department of Physics and Astronomy, University of Padova, 35131 Padova, Italy
2   Department of Mathematics, University of Genova, 16146 Genova, Italy
*   Correspondence: stefania.lippiello@phd.unipd.it (S.L.); boscolo@dima.unige.it (A.B.)

**Abstract:** This study focuses on interdisciplinary approaches within mathematics and physics education. Secondary schools, particularly those specialized in scientific curricula, have opportunities to explore common topics between mathematics and physics; however, creating a coherent interdisciplinary educational experience is challenging. Adopting an interdisciplinary perspective when designing learning sequences becomes imperative. The proposed approach harnesses the power of storytelling to engage students, emphasizing the interconnectedness of subjects and humanizing the evolution of scientific ideas. This study investigates the adaptation of the Digital Interactive Storytelling in Mathematics (DIST-M) model for interdisciplinary storytelling learning sequences. It aims to explore how this model, initially developed for mathematics activities in a virtual environment, can be enriched with elements from inquiry-based learning models to integrate the experimental aspects of physics. The research presents a theoretical discussion grounded in the design of a learning sequence centered around the study of light, taking place in a non-virtual environment and approached from an interdisciplinary standpoint. It introduces hypotheses for adapting the DIST-M model to accommodate interdisciplinary storytelling sequences. One involves the incorporation of an additional phase within the DIST-M cycle, dedicated to consolidating, transferring to other contexts, and addressing variations in the concepts explored, proved, and refined in earlier phases.

**Keywords:** interdisciplinarity; DIST-M model; storytelling in STEM disciplines; narratives in STEM education; physics education; mathematics education

## 1. Introduction

Interdisciplinarity is becoming increasingly prominent in educational, research, political, and institutional contexts [1].

In particular, interdisciplinarity in mathematics education refers to an approach that integrates concepts, methods, and perspectives from multiple disciplines within the realm of education to enhance the teaching and learning of mathematics in different contexts. Rather than treating mathematics in an isolated manner, interdisciplinary mathematics education recognizes the interconnectedness of mathematics with other subjects and real-world contexts. This approach aims to provide a more holistic and meaningful learning experience for students [2].

Interdisciplinarity can be implemented in various ways; some characteristics of interdisciplinary mathematics education may include real-world applications or problem-solving scenarios, but also a deeper integration of different subjects to see the interconnected nature of knowledge and deepen students' understanding [3].

With this approach, it is also possible to make mathematics more engaging, relevant, and accessible to students while preparing them for the complexities of the real world [4].

In the context of science education, the family resemblance approach (FRA) [5,6] is a useful framework for characterizing disciplinary identities and, at the same time,

connecting disciplines by fostering mechanisms of crossing and transgressing boundaries. In the family resemblance approach applied to science, the emphasis is on recognizing the diverse and interconnected nature of scientific practices, methods, and knowledge. This perspective encourages educators to present science as a dynamic and evolving enterprise with different branches, methods, and approaches, thus integrating a reflection on the nature of science in education. It also emphasizes the importance of understanding the relationships and connections between the various scientific disciplines, promoting a more holistic view of science in education. Irzik and Nola's work [7] suggests that adopting a family resemblance approach in science education can contribute to a more complete and realistic portrayal of the nature of science, fostering a deeper appreciation for the interconnectedness and diversity inherent in scientific practices.

In our work, we focus on the theme of interdisciplinarity between mathematics and physics in science education. The relation between the two subjects is often underestimated: mathematics is seen as a tool in physics education, and physics is viewed as a context for applying mathematical concepts, but recent approaches and models are explored to overcome this dichotomy and emphasize the interplay between mathematics and physics [8–13], from a cognitive point of view [8] and also regarding a historical-epistemological perspective that sees them intertwined [13].

Particularly in secondary schools, the challenge of interdisciplinary education involving mathematics and physics is of main concern. In these school grades, especially within schools that address a curriculum focused on scientific subjects, these two subjects have several aspects that intersect and overlap. However, often the discussion of topics relevant to both subjects may occur in a non-cohesive manner within the two, potentially affecting the opportunity to respect the interplay of the two disciplines from an interdisciplinary perspective. For this reason, designing educational sequences from an interdisciplinary perspective appears to be necessary.

A context that allows working on the strong interplay between mathematics and physics is that of modelling [14]. A modelling cycle explicitly framed to effectively incorporate mathematics into physics education is the one proposed by Uhden et al. [15]. The cycle consists of a series of steps that, starting from the real world, involve simplification, mathematization, interpretation, technical mathematical operations, and, finally, validation of the model thus created to describe a real-world phenomenon or problem. The peculiarity of this model lies in the distinction between the technical and structural role of mathematics in physics and, consequently, the related competencies, technical and structural. Technical competencies are related to computational manipulations in a purely mathematical context and concern purely mathematical skills, while structural competencies are related to the ways of reasoning that mathematics provides for understanding physical situations. The key processes that the model describes are mathematization, which consists of formalizing a physical problem with gradually increasing degrees of mathematization; interpretation, which allows physical meaning to be deduced from equations, identifying special cases or making physical predictions from the formalism; and technical-mathematical operations related to purely technical skills.

Since modelling is a relevant competence to be developed in education [16], we can consider it an appropriate context for designing interdisciplinary activities in classroom practice also in secondary school.

Furthermore, to actively engage students in and navigate the challenges of interdisciplinary work, argumentation is seen as an essential competence that enables them to collaborate effectively across disciplinary lines and contribute innovative solutions to challenging problems, as in modelling contexts. Indeed, argumentation facilitates an aware integration of disciplinary and interdisciplinary perspectives through the promotion of critical thinking and the encouragement of communication across different fields of study [5,17]. For the design of interdisciplinary inquiry-based learning sequences that focus on promoting argumentation, employing storytelling could be a promising approach. Indeed, storytelling can be a useful educational tool for emphasizing the interrelationships among

various subjects, tracing their historical evolution, and fostering students' engagement and active participation [18]. Since narrative thinking [19] is a powerful method for infusing meaning into our experience, the process of creating stories is an effective means of conveying the meaning of concepts, models and theories in the scientific domain, especially when the scientific process becomes the direct experience of the character within the story [20]. This approach helps to draw parallels between the art of storytelling and the practice of systems modelling in science, as noted by [21]. In particular, in the realm of mathematics, storytelling takes on added significance when narrative thinking is developed in synergy with logical thinking [22].

As an emergent field of research, there is no explicit reference model to design interdisciplinary activity between mathematics and physics framed upon storytelling. On the contrary, to develop a storytelling mathematics-related activity centered on the argumentative competence, a reference model for instructional design is Digital Interactive Storytelling in Mathematics (DIST-M) [23–25]. This model was originally created for developing mathematics activities in a virtual environment. However, more generally, it is possible to consider it a tool for designing storytelling mathematics learning sequences, even if the digital component is not included. Then, DIST-M can be considered as a starting model to frame the learning sequence. Nevertheless, to foster an interdisciplinary approach between mathematics and physics that upholds the significance of both disciplines, it is essential also to consider the experimental features inherent to physics [5]. Specifically, research shows that it is particularly effective to introduce laboratories in education with objectives aimed at developing scientific practices/skills [26,27]. Moreover, in the 2012 Framework for K-12 Science Education [28] we can find 'Planning and carrying out a systematic investigation' as a major practice of scientists that should be strengthened throughout the K-12 curriculum.

Looking at the experimental aspect, an Inquiry-based learning approach [29], to which, for example, Kolb's model [30,31], 5E [32], and the Investigative Science Learning Environment (ISLE) refer [33–35], can guide the design choice to give relevance to these core components of physics. Thus, the following research question emerges: *How and to what extent can the experimental features emphasized by design models of inquiry-based learning be integrated within the storytelling model proposed by DIST-M?*

This study aims to explore how to effectively integrate an interdisciplinary approach between mathematics and physics within the realm of storytelling. In particular, what adaptations the DIST-M model, originally intended to develop mathematics activity, may require to be a design model for a storytelling interdisciplinary learning sequence? In this contribution, we will present a theoretical dissertation derived from the example of the design of a learning sequence concerning the study of light, from an interdisciplinary perspective. This learning sequence emerges from the collaborative efforts of educators and researchers within the paradigm of action research, aiming to design an interdisciplinary instructional path focused on developing modelling and argumentation skills [36]. Subsequently, we will propose hypotheses for the adaptation of the DIST-M model for developing interdisciplinary activities from this initial example. This adaptation will be based on the design models for Inquiry-based learning, considered in the development of the sequence to enhance the experimental component that distinguishes the field of physics. This learning sequence is provided exclusively as an example for theoretical discussion purposes. This contribution will not deepen the aspects related to its implementation in school contexts and justifications of the decision in this direction (e.g., students' participation, group work, assessment) will be not illustrated here. However, future research will move in the direction of examining the outcomes of the learning sequence implementations.

## 2. Models for Instructional Design

In this paragraph, we will present the fundamental models we refer to in our dissertation. The models were chosen with reference to our target objectives: to work in an interdisciplinary manner through modelling by promoting argumentation. To develop an interdisciplinary teaching–learning sequence on storytelling, we start framing with the

reference model for designing storytelling activities in mathematics DIST-M [23–25]. From this starting point, to integrate other guides to encompass the physical dimension, from an experimental perspective, we consider Kolb's model [30,31]. What resulted was not a simple overlap of approaches, but involved careful reflection on the role of each model for the individual discipline and the interconnection between disciplines. This reflection was guided by the Family Resemblance Approach (FRA), for keeping in mind the core features of the involved disciplines, and the work of Uhden et al. for the attention on modeling between mathematics and physics.

### 2.1. The DIST-M Model

DIST-M guided us to design a compelling narrative structure that seamlessly integrates disciplinary concepts.

As general directions, according to the model, the story must provide context and relevance to the content, creating a cohesive and engaging learning experience. At the same time, the narrative must allow students to actively participate by encountering problem-solving included in the storyline and relevant to its progression. This encourages critical thinking and reinforces the use of mathematics in real contexts. The model highlights that the experience must, as far as possible, be customizable to different learning rhythms and assessment must also be integrated into the storytelling experience. On the one hand, design choices must promote collaborative learning, fostering shared work and argumentation skills. On the other hand, providing feedback to students within the storytelling environment is considered essential. Feedback helps guide students, reinforcing deep and meaningful understanding of concepts.

Concerning the structure of a storytelling learning sequence framed upon the DIST-M model, the following steps are identified [23,24]:

- Phase 1—*inquiry*: students begin to explore the problem, investigate the hypothesis leading to an initial and personal conjecture (even if only verbal).
- Phase 2—*conjecture and formalization*: students discuss and manipulate the initial statements to achieve a formalized one.
- Phase 3—*arguing and proof*: students, then, attempt to prove the conjecture, justifying each step of the deduction.
- Phase 4 and 5—*summing up and refining*: students, when retelling a story, reflect on the entire process that led to the solution of the problem. This step helps to evaluate the work done and the role played (self-assessment, metacognitive, and affective level).

Other descriptions of the cycle have emerged in previous works related to the DIST-M model [25], focusing on the articulation stages of five episodes within the story, aiming to support the argumentative process: *Exploration, Conjecture, Formalization, Proof*, and *Reflection*. Apart from a re-arrangement of the phases, the overall structure of the cycle remains identical to the previous one. Specifically, we can observe that phases 4 and 5 have been further condensed into the final phase. It justifies the grouping we made in the previous list, considering them within a single step.

### 2.2. The Inquiry-Based Learning Models

In the field of inquiry-based learning, some planning structures help teachers develop student-centered inquiry-based lessons and units, such as the Kolb [30,31] or 5E [32] model. We have chosen the former to create a more complete and experiential learning environment that suits different students with their different learning preferences. The cyclical nature of the Kolb model emphasizes the importance of continuous reflection and application in the learning process. To summarize, Kolb's model is a cyclic process involving four stages (Figure 1):

1. *Concrete experience*, which involves direct, practical experiences as the starting point of the learning process.
2. *Reflective observation*, which promotes a reflective attitude on what has been observed to encourage the formulation of questions and the search for answers.

3. *Abstract conceptualization*, in which students analyze their observations and reflections to generalize, move towards abstract concepts and finally develop laws and theories.

4. *Active experimentation*, which involves applying concepts and theories to new situations or actively testing what has been learned. This practical experimentation completes the learning cycle and prepares the learner for the next concrete experience.

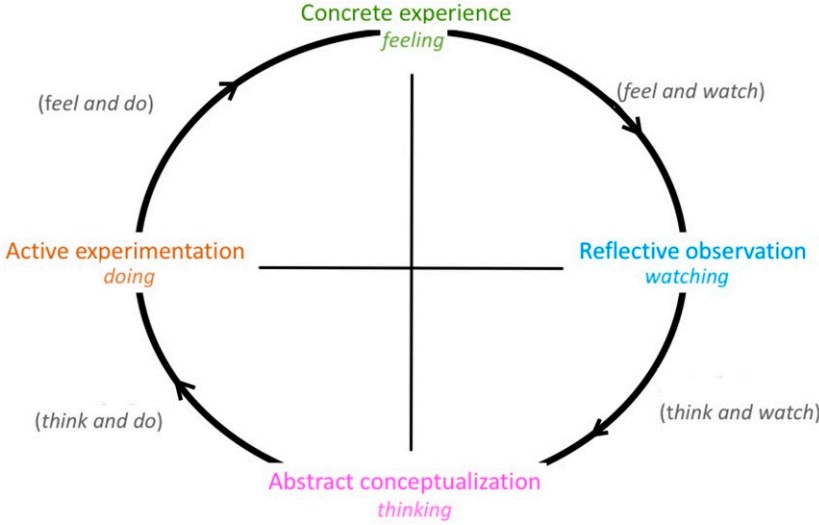

**Figure 1.** Schema of the Kolb cycle.

Especially focusing on the role of experimental experience, a particularly useful approach for didactic implementation is the Investigative Science Learning Environment (ISLE) because it has an emphasis on aspects of didactic meta-reflection, while also providing a range of validated teaching materials, including assessment rubrics that encourage the development of scientific practices. It can be used to support students in learning physics by involving them in processes that reflect scientific practice. The ISLE model can inspire the use of different kinds of experiments (observational, testing, application), the structure of the laboratory worksheets, and the assessment rubrics (in addition to what has already been stated in the previous paragraph) as in [33–35].

### 3. Research Design

For discussing the adaptation of the DIST-M model for a storytelling interdisciplinary mathematics–physics learning sequence, we make use of a first example: the design of a learning sequence concerning the study of light, aimed at developing argumentative and modeling competence.

As a reference to pinpoint the core characteristics of an interdisciplinary learning sequence, the FRA appears effective in respecting the epistemological complexity of interdisciplinarity and in providing categories (aims and values, practices, methods, and methodological rules, knowledge organized between cognitive-epistemic and a social-institutional system) to reason about disciplinary identities and their dialogue. The FRA framework thus allows reflection on what characterizes the scientific disciplines and, at the same time, promotes explicit disciplinary meta-reflections on the epistemologies of mathematics and physics. Under its guidance, we can, therefore employ Uhden's model, which delves into the role of mathematics in physics, providing a model for emphasizing the translation process between physics and mathematics. In particular, simplification and validation connect the world with the physical model, while mathematization and interpretation connect the physical model with mathematics. Indeed, Uhden's model can be seen as potentially asymmetrical, since the focus is more on physics and the role of mathematics in physics [37]. We thus consider its involvement appropriate to balance the fact that our starting point on interdisciplinarity is from the mathematical field. Even though it is usually considered as a framework to model a problem, in this work, it is

conceived as a framework guiding the design of the learning unit itself, further than being taken into account to structure the various activities.

In particular, the theoretical characterization of Uhden et al.'s work highlights the need for integrating a supplementary framework to design the learning unit, further than the DIST-M reference. Indeed, we start conceiving the DIST-M model, which fits since the activity also involves mathematics, but we further need to integrate aspects of Kolb's design, to address the characterization of physics as having an experimental nature.

The activity is designed as described in the following section. Starting from the example provided, a possible adaptation of the DIST-M model to the interdisciplinary learning sequence will be discussed in Section 4, looking back to the design of the unit from the perspective of Kolb and the DIST-M cycle.

*The Learning Sequence*

The developed learning sequence aims to study light, a central topic in physics that lends itself to interdisciplinary reading, particularly with mathematics, and also promotes awareness of the nature of science. The unit revolves around three core questions about light: *What is the nature of light? How does it propagate? How does it interact with matter?* To address these questions, specific topics covered are the models of light—rays, particles, and waves models—and the first phenomena concerning the interaction of light and matter: reflection and refraction.

The learning objectives of the learning sequence are inherently interdisciplinary. The students are expected to have a comprehensive understanding of the models of light and should be able to explain various phenomena involving light-matter interaction and predict the outcome of experiments. It includes using mathematics to identify, compare and generalize, formulate laws (geometry, sine function, and basic algebra), and solve problems in optics. Other objectives align with the choice of making use of storytelling in the learning sequence: the students should be able to collaborate with peers to find solutions to problems, and actively participate in discussions being responsible for expressing their own thinking and contributing to the problem-solving process.

The learning sequence has been implemented in a class of 14/15-year-old students (grade 9) in a scientific-oriented high school with an experimentation involving four hours per week for 6 weeks. Adapted to the specific context of the class, including the students' interests and dynamics, the intervention employs a narrative approach inspired by "Lord of the Rings". The class comprises 18 students and, although they show interest, few actively engage in discussions. To solve this problem, it was decided to implement storytelling and role-playing in order to increase participation and feedback to improve learning and attitude.

The unit has been developed starting from the DIST-M model, organizing the narratives in multiple cycles: a macro-cycle concerning an overall view on the main questions, which starts in the introductory phase and continues in the concluding part of the sequence, and two other DIST-M cycle in between, the first explicitly referring to reflection and the second to refraction (Figure 2).

Each cycle begins with the *Inquiry and Conjecture* activities (points 1 and 2 of the above list), continues with the tasks and discussion on *Arguing and Proof* (point 3), and ends with the *Summing up and Refining* section (points 4 and 5). More precisely, a lesson explicitly aimed at phases 4 and 5 has been included in the concluding part of the macro-cycle, which involves the overall contents of the sequence. However, a partial Summing up and Refining phase has been integrated into each cycle.

Apart from the narrative script and didactic structure that follows the DIST-M cycle, as mentioned above, the overall design of the learning sequence was developed according to the characteristics of the DIST-M model as well, except for the digital aspect. We start the learning sequence with an explorative problem, which has to be solved dealing with several activities. The focus of the entire sequence is on the collaboration, and the students (divided into groups) and the expert (teacher or researcher) playing clearly defined roles

integrated into the narrative. The story evolves precisely according to the interactions between the characters and the stimuli coming from the expert. Furthermore, evaluation takes place through both collective and individual feedback, which is constant within the story. Other formative assessment strategies were included, such as self-evaluation, often related to the partial Summing up and Refining phases. Traces can be found in the quest provided in Appendix A (Figures A4, A5, A7 and A10).

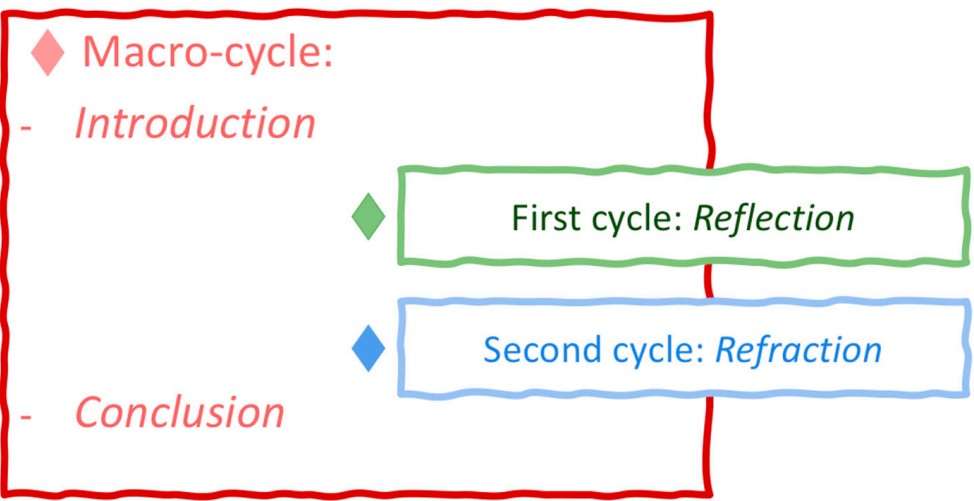

**Figure 2.** Schema of the three DIST-M cycles in the learning sequence.

The narration upon which the story is articulated is inspired by "The Fellowship of the Ring" by Tolkien. To prioritize the engagement, the narrative framework is chosen according to teachers and students' interest. Thus, in a different context, another background might be a better choice. Through the narration, students start a journey and delve into a series of adventures. The experience will give them the expertise and knowledge necessary to reach the final goal of the story, which is aligned with the learning objects of the learning sequence.

The story begins with Galadriel giving Frodo a crystal vial containing the light of the star of Eärendil to help him in his important task. A new group of characters arrive in Lórien and are asked to help the Fellowship against the dark forces of Sauron, gaining experience through quests. These are the students, who become the characters of the story with unique abilities based on Tolkien's races. During their journey, they encounter various challenges related to light and its interaction with matter, exploring concepts such as reflection and refraction and trying to understand the nature of light through multiple models (ray, wave, particle). This knowledge will be needed to solve the problems they encounter, for example, to repel orcs and detect poison in the blood of elves using light, and will enable them to really help the Fellowship.

Without going into too much detail, but in order to illustrate the overall structure of the learning sequence, in Table 1 we explain the modules that make it up through the learning objectives and questions that guided its development. In Table 1, the refraction cycle is not included because we will analyze the cycle in detail later, showing illustratively how the cycles were created.

In the following, we will illustrate how the refraction cycle has been designed, starting from the narrative framed upon the DIST-M model, after having identified the core theme and learning objective to be addressed. This provides an example of how the whole learning sequence has been created upon the selected models, under the guidance of Uhden's directions.

In Table 2, the script of the narration upon which the refraction cycle is articulated, framed on the DIST-M model phases, is briefly described. In the last column on the right,

there are indicated the didactic modules related to the specific narrative phases. A short description of each didactic module is provided in Table 4.

**Table 1.** The learning sequence described through the objectives and the guiding questions.

| The Learning Sequence | Objectives | Guiding Questions |
|---|---|---|
| Module 1 *Introduction* | Bring out the initial knowledge possessed. Start observing and asking questions | What is light? What do you know about light? |
| Module 2 *Ray model* | Understand how light travels, what is needed for us to see, and if light interacts with matter. | How does light travel? How/why do we see an object? |
| Module 3 *Wave and particle models* | Understand if the ray model of propagation is consistent with the wave or particle model. | What is the nature of light? What is light made of? Is it made of waves or particles? |
| Module 1 *Observational experiment* | Experience and experiment reflection in different situations | How does light interact with matter? |
| Module 2 *Reflection within models* | Explore the specular reflection within the different models | What is the nature of light? Are the reflection phenomena evident within the models? |
| Module 3 *The law of reflection* | Devise a rule for specular reflection | How can we formalize the previous observations? Can we infer a law? |
| Module 4 *Applications* | Solve problems applying the reflection law, and experiment reflection phenomena in different situations (e.g., curved mirror) | Can we apply the law of reflection in different situations? |
| Refraction Cycle (in Table 4) | . . . | . . . |
| Module 1 *Recap* | Reorganize and reorder concepts, fix ideas, improve and evaluate learning. | So what have we learnt so far? |
| Module 2 *Game solution* | Apply the knowledge acquired to new situations | So how can we use what we have learnt so far? |

**Table 2.** Refraction cycle: Narration articulated on the DIST-M phases.

| DIST-M | Narrative Script | Didactic Modules of Refraction |
|---|---|---|
| Phase 1 *Inquiry* | They eat breakfast and watch a spoon in the water, so they say that when they washed they saw their feet in the basin, they looked strange. Other characters say that they dropped the soap in the water and to pick it up it was in a different place from where it seemed to them. . . Our heroes finally reach the Fellowship! But they had to protect themselves with a strange technique: the entrance of the gate seemed frozen in a sort of substance similar to glass. . . They can see the stairs and a plate! It's a code. . . Only the worthy can enter. After decoding the message, they understand that if they can hit the switch with the light, they will be able to enter. But they only get one try. | Module 1 *Explore refraction physically* School trip *Visit to the museum Poleni (Padova)* |
| Phase 2: *Conjecture* | They go back and go to a glass artisan so they can do experiments in order to understand how the law of this phenomenon works. | Module 2 *Explore refraction mathematically* |
| Phase 3: *Proof* | Not understanding the regularity, they go to the library and find writings in human language; only humans can read them. They study the sine function and they solve the problem theoretically. | Module 3 *The law of refraction* |
| Phase 4 and 5: *Summing Up and Refining* | It is important that the law is correct, they have only one attempt; otherwise the mission will fail. | Module 3 *The law of refraction* |

Having reached phase 4 and 5, the DIST-M cycle would be concluded. However, in order to emphasize the inherent aspects of physics within the interdisciplinary learning sequence, recourse is again made to FRA and Uhden's model, and the essential role of the experimental and applied nature of physics. Putting attention on this component, to help us guide the analysis of how the experimental components fit in the sequence, we will refer to Kolb's model.

**Table 3.** Refraction cycle: The additional phase.

| DIST-M | Narrative Script | Didactic Modules of Refraction |
|---|---|---|
| - | They still check, trying to apply the law to see if it predicts well the behavior of light when it passes through transparent materials. They also check with the models by talking to Huygens and Newton... They have to be really sure before they exploit their attempt. And it works! They manage to reach the Fellowship. | Module 4 *Application of the law of refraction* Module 5 *Refraction and models* Module 6 *Solving the problem* |

Adopting DIST-M, which is a framework for developing mathematics activities, the cycle is structured around phases that correspond to the mathematical components of the unit. However, with adaptation concerning how physics is taken into consideration, the sequence also aligns with part of Kolb's cycle. Indeed, the first three phases of the Kolb cycle could be traced back to the designed modules, in parallel with the DIST-M cycle: the *concrete experience* in Module 1 and in the school trip, as well as the *reflective observation* in Module 2 and part of Module 1, and *abstract conceptualization* in Module 3. However, the last phase of the Kolb cycle, corresponding to *active experimentation,* cannot find a place within the planned didactic modules. This phase involves the application of the concepts and laws derived to new situations or the active verification of what has been learned.

Therefore, in line with the Kolb cycle, we expand the narrative and didactic sequence with a further phase, which we refer to as *Phase 6*, in which we incorporate didactic modules regarding the reflection concerning the re-interpretation within the physical models of what has been discovered in terms of mathematical laws, and the applications, and the verifications of these laws in further problematic and experimental situations (Table 3).

In Table 4, the whole structure of the Refraction cycle, articulated in didactic modules, is illustrated. Each module has been associated with the specific learning objective addressed, the questions that guide the module, the typology of activities involved in the module, and features concerning how the module is planned to be implemented in class. The worksheets guided the requests encountered by the students in the development of the story, as quests to be completed. The quests related to the cycle here described are illustrated in Appendix A.

**Table 4.** Refraction cycle: Descriptions of didactic modules.

| Cycle 2 Refraction | Objectives | Guiding Questions | Type of Activities | Didactical Aspects of the Implementation |
|---|---|---|---|---|
| Module 1 *Explore refraction physically* | Experience and experiment refraction in different situations | How does light interact with transparent matter? | Observational experiment Data collection (qualitative) | Classroom organization: Groups of 3 students Didactical materials: Quest 6 Refraction (Figure A1) Expected duration: 1 h |
| Module 2 *Explore refraction mathematically* | 1. Devise (discover) a rule for refraction 2. Search for regularities (conjecture) | Do you notice some regularities or not? Which mathematical relations do you know? Are they useful here? | Data collection (quantitative) Data analysis | Classroom organization: Group work Didactical materials: Quest 6—Refraction (Figure A2) Expected duration: 1 h |

**Table 4.** *Cont.*

| Cycle 2 Refraction | Objectives | Guiding Questions | Type of Activities | Didactical Aspects of the Implementation |
|---|---|---|---|---|
| School trip *Visit to the museum Poleni (Padova)* | 1. Deepen the nature of science through history 2. Observe an instrument asking questions | What is the nature of science? What relationship do you see between the discoveries shown at the museum and what we still need to understand about light? | Object-based learning [38] Observe an instrument chosen between the burning mirror and refractometer and ask a minimum of 30 questions about it. | Quest 7—A journey (Figures A6 and A7) |
| Module 3 *The law of refraction* | 1. Deepen Mathematics: the sine function 2. Find a rule for refraction 3. Formalize the law of refraction | Now, that you have the new instrument of sin: what have you discovered in your observations? Do you notice some regularities or not? Can you build up a rule? | Mathematical stage to learn the sin-machine Arguing conjectures Collective discussion Come into proof | Classroom organization: Groups of 3 students and collective discussion Didactical materials: Documents about the sin-machine (Figure A7) Quest 6—Refraction (Figures A3–A5) Expected duration: 2 h |
| Module 4 *Application of the law of refraction* | Solve problems, applying the law of refraction to different situations | How can we apply the law of refraction? | Problems (including a jeopardy problem) Recognizing refraction in everyday life Observational experiment with convex and concave lenses | Classroom organization: Groups of 3 students Didactical materials: Quest 8—Refraction applications (Figure A8) Expected duration: 3 h |
| Module 5 *Refraction and models* | 1. Check if the models are consistent with the rule for specular refraction 2. Discuss the historical positions of Newton and Huygens on the wave and particle models | Are the models consistent with the phenomenon? What is the model of light that seems to describe best what we have discovered about light? | Experimentally, try to understand if the law of reflection is consistent with the models Read a historical article about light Overall discussion and conclusions | Classroom organization: 3 groups of 3 students of the wave faction, and 3 groups of 3 students of the particle faction Observing and Reading in groups Whole class discussion Didactical materials: Passages from the book *The Evolution of Physics* by Einstein and Infeld [39] Expected duration: 2 h |
| Module 6 *Solving the problem* | Solve a contextualized open problem by putting into practice the whole knowledge reached about refraction | How to creatively interpret a phenomenon and apply refraction to solve contextualized problems? | Problem-solving with group discussion Negotiation to find a common, agreed solution | Classroom organization: Mixed groups (3 for groups) Didactical materials: Quest 9 (Figures A9 and A10) |

## 4. An Exemplary Integration of Models from an Interdisciplinary Perspective

By using DIST-M to develop an interdisciplinary learning sequence between mathematics and physics with an experimental connotation, it becomes evident from the example presented in the previous paragraph (Tables 2 and 3) that it is necessary to add a phase. Indeed, in the experimental domain, it is essential to repeat experiments by applying them to new situations in order to continue testing their validity [33,34].

What was presented in the exemplar cycle occurred for the entire sequence, in each of its cycles. In the previous section, we showed it for the refraction cycle, but, as can be seen in the summary table below (Table 5), it occurs systematically within all cycles and thus for the entire learning sequence.

In the DIST-M model, the last phase is intended as a form of evaluation. After the students have formalized the proof, organizing and justifying the deductive steps, we then move on to a reflection phase that is useful both as a form of collective and self-assessment, while remaining on a cognitive, metacognitive, and affective level. Instead, in the field of experimental physics, we need a further moment in which to apply the concepts to new situations, experimentally and actively verifying what we have learnt. This is, for example, what Kolb's model's 'Active Experimentation' phase provides for. On the other hand, DIST-M helps us to integrate the structural role of mathematics into physics, focusing on reasoning, formalization, and proof.

Intending to follow Uhden's suggestions as a design guide, integrating DIST-M and Kolb models could generate a complete model for designing experimental interdisciplinary activity. Then, we propose to adapt the DIST-M model, adding the phase called *Consolidation, Transfer, and Variation*. It serves to consolidate what has been learned, apply it in a new context, and understand how it can be enriched by this, and what changes it can undergo.

This phase thus encompasses problem-solving of various kinds and new experiments (e.g., tests or applications according to ISLE approach [33–35]) that validate or modify what has been theorized.

**Table 5.** The learning sequence declined in modules and analyzed according to DIST-M and Kolb model, with reference to the guidance of Uhden's framework.

| The Learning Sequence | | DIST-M Model | KOLB Model |
|---|---|---|---|
| **Introduction (macro-cycle)** | Module 1 *Introduction* | Phase 1: Inquiry | Concrete experience |
| | Module 2 *Ray model* | Phase 2: Conjecture Phase 3: Arguing and Proof | Reflective observation Abstract conceptualization |
| | Module 3 *Wave and particle models* | Phase 2: Conjecture Phase 3: Arguing and Proof | Reflective observation Abstract conceptualization |
| **Reflection cycle** | Module 1 *Observational experiment* | Phase 1: Inquiry Phase 2: Conjecture | Concrete experience Reflective observation |
| | Module 2 *Reflection within models* | Phase 1: Inquiry Phase 2: Conjecture | Concrete experience Reflective observation |
| | Module 3 *The law of reflection* | Phase 3: Arguing and Proof Phase 4 and 5: Summing Up and Refining | Abstract conceptualization |
| | Module 4 *Applications* | Phase 6: Consolidation/Transfer/Variation | Active experimentation |
| **Refraction cycle** | Module 1 *Explore refraction physically* School trip *Visit to the museum Poleni (Padova)* | Phase 1: Inquiry Phase 2: Conjecture | Concrete experience Reflective observation |
| | Module 2 *Explore refraction mathematically* | Phase 2: Conjecture | Reflective observation |
| | Module 3 *The law of refraction* | Phase 3: Arguing and Proof Phase 4 and 5: Summing Up and Refining | Abstract conceptualization |
| | Module 4 *Application of the law of refraction* | Phase 6: Consolidation/Transfer/Variation | Active experimentation |
| | Module 5 *Refraction and models* | Phase 6: Consolidation/Transfer/Variation | Active experimentation |
| | Module 6 *Solving the problem* | Phase 6: Consolidation/Transfer/Variation | Active experimentation |
| **Conclusion (macro-cycle)** | Module 1 *Recap* | Phase 4 and 5: Summing Up and Refining | Abstract conceptualization |
| | Module 2 *Game solution* | Phase 6: Consolidation/Transfer/Variation | Active experimentation |

The DIST-M integrated with the new phase at the end becomes consistent with Kolb's model and with what Uhden advocates. We can trace back the steps of the Uhden model, adapted to be read as design guidance, in all the cycles that make up our learning sequence, as well as within a single module. For example, looking at the macro-cycle, we can find in the Introduction a *simplification* phase followed by *mathematization*, when we start using geometry, and then return to physics by *interpreting* what we have obtained mathematically. Analyzing the wave and particle models, we start again with *simplification, mathematization,* and *interpretation*. Then, the macro-cycle concludes with the *validation* phase and, when applied to the solution of the game, all the phases of Uhden's model return. This is true also for internal cycles. For instance, looking at the refraction cycle, we can find in the Module 1 a *simplification* phase followed by *mathematization* in Module 2, when we start searching mathematical relations and formalizing, doing also some *technical mathematical operations* to find a law (with the highest degree of mathematization) in Module 3, and then return to physics by *interpreting* what we have obtained mathematically. In Module 4 applying the law to new different situations involves the *validation* phase, but also inside

the various proposed activities the Uhden cycle is retraced again. In Module 5 checking if the models are consistent with the rule for specular refraction is also a *validation* stage. Finally, the solution of the contextualized open problem serves both as the *validation* phase of the entire process, but also internally requires all steps of the Uhden cycle again.

## 5. Discussion and Conclusions

To adopt an interdisciplinary approach that integrates mathematics and physics, first of all, we considered the Family Resemblance Approach [5–7] to always keep in mind aims and values, practices, and knowledge that distinguish one disciplinary domain from another, particularly for what characterizes science [5].

In parallel, to treat the relationship between mathematics and physics and find the right way to integrate and interlace them, we propose to draw on the theoretical guidelines provided by Uhden and colleagues [15]. We view their theorization as a valuable resource for guiding the design of a learning sequence that originates from an interdisciplinary perspective. It incorporates the intertwining of mathematics and physics and also allows us to distinguish between technical and structural skills, bearing in mind that the role of pure mathematics and pure qualitative reasoning should not be neglected, but resolving the translation process between physics and mathematics in more detail. This model is also useful in highlighting different ways of reasoning, highlighting some possible sources of difficulty, and suggesting more appropriate approaches. In contrast to the common reference to Uhden et al.'s theorization for analyzing students' thinking or for planning individual math-physics activities, the perspective adopted in the research goes further. We see this perspective as a framework to be considered when planning an entire learning sequence. However, it is important to note that Uhden's work, while offering guidance for the planning of specific activities, does not by itself constitute a truly comprehensive instructional design model. For this reason, we believe that it is necessary to supplement these guidelines with models explicitly created for the design of learning sequences.

In the specific case of designing an interdisciplinary math-physics learning sequence centered around storytelling, we found it essential to refer to a specific design model suited for this type of activity. In the absence of an interdisciplinary model specifically directed at designing mathematics and physics storytelling sequences, we turned to the well-established DIST-M model [23–25] within the field of mathematics. This model is considered suitable for creating argumentative mathematics learning sequences, and, even if it originates in virtual contexts, it is enough feasible to be extended in non-digital environments. Indeed, DIST-M provides a structured framework with specific characteristics, such as the focus on Inquiry–Conjecture–Proof progression, collaborative, role-driven work, continuous feedback for monitoring learning, and guided reflection aspects for developing metacognitive skills, that can be adapted for use outside of digital contexts.

However, following the indications of the FRA and Uhden, it became evident that the DIST-M model, being designed for the mathematical domain only, did not incorporate the necessary components to comply with the aims and values of physics. In particular, the experimental aspect was underestimated within this model, and the reinterpretation of mathematical results within the physical context and applications, as well as the interpretation phase of the physical meaning of mathematical expressions and the possibility of making physical predictions from the formalism was lacking. Consequently, the need arose to supplement this model with a design framework highlighting the fundamental steps of an experimental teaching unit focusing on inquiry-based learning.

In this context, we chose to incorporate Kolb's model [30,31] for designing experimental activities as an integral part of the learning sequence and, consequently, the narrative script. This adaptation forced us to extend the DIST-M model by adding a phase beyond the existing model, emphasizing the features highlighted by Uhden which we recall consists of the attention to the structural role of mathematics in physics, expressed by the phases of mathematization, which consists of formalizing a physical problem with different degrees

of mathematization, and interpretation, which allows the physical meaning to be deduced from the formal mathematical language.

In this contribution, we illustrated an exemplary case to support our argument that an integrated design model should be considered when designing interdisciplinary teaching sequences based on storytelling. We highlight, starting from this case, the necessity of adapting the DIST-M model by revisiting the design structure, revaluating Uhden's model as a guiding framework for the didactic sequence, and comparing the cycles with those proposed by Kolb.

As one possible adaptation, we suggest extending the DIST-M cycle with an additional phase focused on consolidation, transfer, and variation in what has been learned in the previous phases. This adaptation leads to the creation of an integrated model, which we propose to call "Interdisciplinary Interactive Storytelling in Mathematics and Physics" (IIST-MP). This model envisions a design script for storytelling in interdisciplinary activities comprising five phases: those outlined by DIST-M, along with the new sixth phase aimed at consolidation, transfer, and variation (Figure 3).

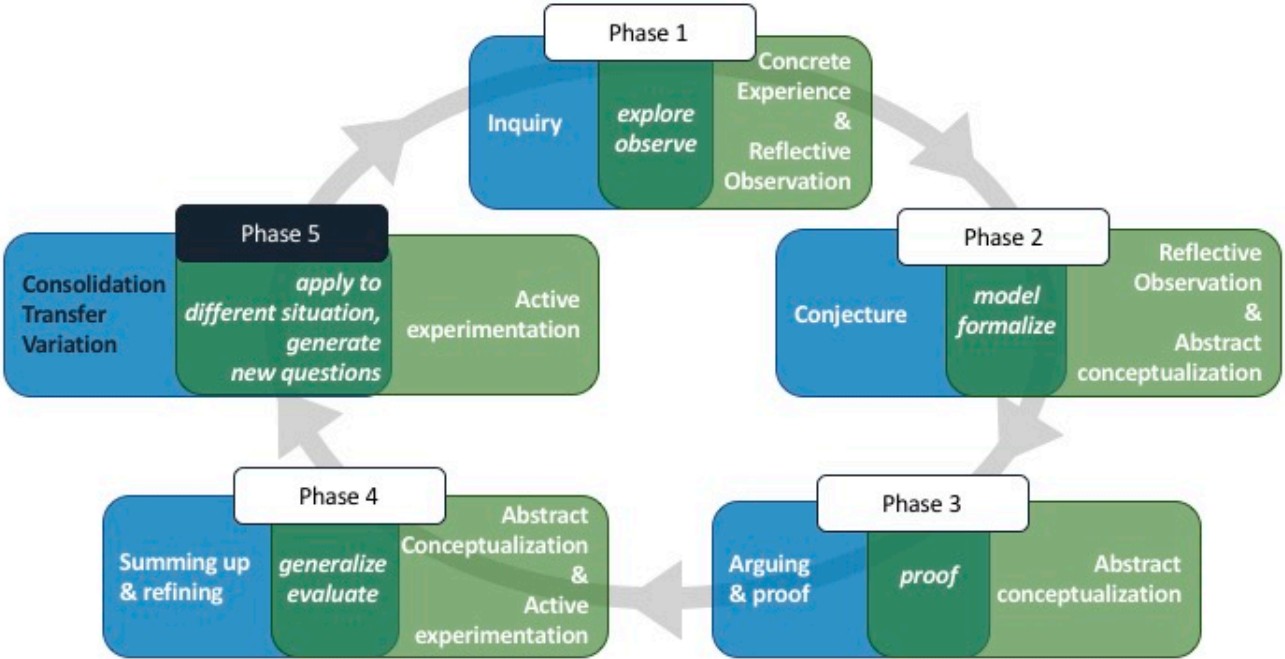

**Figure 3.** Schema of the IIST-MP cycle.

Furthermore, it is essential to emphasize that the adaptation goes beyond surface changes. Within each phase, we should also expand the perspective to include relevant aspects of experimental characterization, as outlined in the Kolb model, following Uhden's guidance. This entails revisiting the steps of DIST-M by incorporating a physics reinterpretation and considering each phase from Uhden's perspective.

## 6. Limitation and Further Directions

In this article, we discussed the need to adapt the DIST-M model to design interdisciplinary experimental activities in mathematics and physics, with a focus on storytelling. We achieved this by examining the necessity of integrating experimental design models, especially the one proposed by Kolb. To illustrate this, we used a specific example related to an argumentative and modeling activity centered around light. We found this example particularly valuable as it showcased how the Kolb model required adjustments through multiple cycles. However, we specifically detailed the educational design process in relation to one cycle, that of refraction. The other cycles are exclusively briefly illustrated in Table 1 showing the didactic modules.

While our discussion was based on a specific context, which we considered significant, it necessitates further investigation to validate the hypotheses of adaptations derived from our study. Additionally, for the development of an interdisciplinary storytelling activity in the realm of mathematics and physics, we should attempt to apply the IIST-MP design model here outlined to a different topic, narrative and context. A further direction may be to investigate if this model can be considered to design interdisciplinary activities involving mathematics and other experimental sciences within the STEM field further than physics. This reflection also extends to our choice of Uhden as a guiding framework for interdisciplinary design. Indeed, in our work, we adopted Uhden's theorization as a design guide, because it appears to provide a solid perspective from an interdisciplinary standpoint, aligning with the FRA framework, as advocated by Satanassi et al. [37]. Nevertheless, utilizing this theoretical model as a guide for designing interdisciplinary activities warrants further exploration.

Our reflection on the integration of models also raises questions about whether selecting the Kolb model for the design of experimental activity is the most suitable approach. Specifically, we ponder if adopting a different model, such as the 5E model [32], would yield similar requirements for adapting DIST-M or result in different outcomes.

Finally, in our contribution, we used the DIST-M model, although we did not consider its original feature of being designed for virtual activities. A possible future direction could be to explore the integration of the DIST-M model from an interdisciplinary perspective, in cases where it is employed for an activity developed in a digital environment, as originally conceived.

**Author Contributions:** Conceptualization, S.L. and A.B.; Methodology, S.L. and A.B.; Validation, S.L. and A.B.; Formal analysis, S.L. and A.B.; Investigation, S.L. and A.B.; Data curation, S.L. and A.B.; Writing—original draft preparation, S.L. and A.B.; Writing—review and editing, S.L. and A.B.; Visualization, S.L. and A.B.; Supervision, S.L. and A.B.; Project administration, S.L. and A.B. All authors have read and agreed to the published version of the manuscript.

**Funding:** This research received no external funding. The APC was funded by the MDPI Editorial Office.

**Institutional Review Board Statement:** This study was approved by the Ethics Committee of the University of Genova (Comitato etico per la ricerca di Ateneo, CERA, protocol code n. 2023.66 and date of approval 29 September 2023).

**Informed Consent Statement:** Informed consent was obtained from all subjects involved in this study.

**Data Availability Statement:** The data presented in this study are available on request from the corresponding author.

**Acknowledgments:** The research developed from a collaboration formed during the BrEW Math 01 (Brixen Education Workshop on Storytelling in STEM disciplines at the crossroads of science and humanities) held at the MultiLab of the Faculty of Education, Free University of Bozen-Bolzano, 8–10 November 2022.

**Conflicts of Interest:** The authors declare no conflict of interest.

## Appendix A

Below are reported worksheets given to students for quest 6 (Figures A1–A5), quest 7 (Figures A6 and A7), quest 8 (Figure A8), and quest 9 (Figures A9 and A10).

## Quest 6

Now the question is: *how does light interact with transparent matter?*

Objective: experience and experiment refraction in different situations; devise (discover) a rule for refraction; search for regularities (conjecture) using mathematics

Names of group participants

_________________________________

### *Observational experiment 1*

Goal: to observe what we see when an object is immersed in water.

A container is filled with water and an object is placed inside, what do you observe?

_________________________________________________
_________________________________________________
_________________________________________________
_________________________________________________

### *Observational experiment 2*

Goal: notice the change in direction of light when it passes through the material.

Vary the direction of the light source so that it passes through a transparent material, what do you observe? Describe with your own words.

_________________________________________________
_________________________________________________
_________________________________________________
_________________________________________________

Make a picture.

What changes compared to reflection?

_________________________________________________
_________________________________________________

**Figure A1.** Quest 6_Part 1.

Let us agree on the vocabulary:

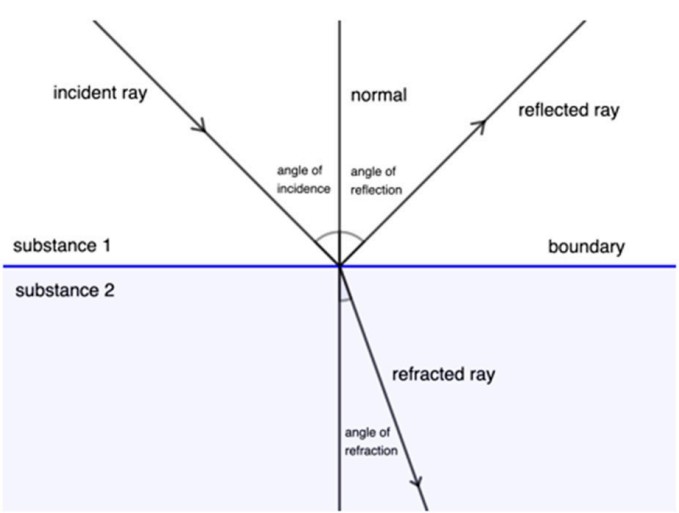

Now repeat the experiment, but collect the data: angle of incidence and corresponding refracted angle. In particular, as the angle of incidence changes, how much is the corresponding refracted angle? Describe the procedure using tables and drawings.

___________________________________________________________

___________________________________________________________

___________________________________________________________

___________________________________________________________

Table the corresponding angles of incidence and refraction.

Do you notice any relationships between those studied? Do they increase or decrease, do they do it directly proportional, inversely proportional, etc.? Test them (use the back of the paper).

___________________________________________________________

___________________________________________________________

___________________________________________________________

Can we find a quantitative relationship as done for reflection?

**Figure A2.** Quest 6_Part 2.

### Observation experiment 3

Goal: to observe what happens when we change medium.

Repeat Experiment 2, but change the material and take note of the results to compare them with those obtained in Experiment 2.

Do this with this simulation:

https://phet.colorado.edu/sims/html/bending-light/latest/bending-light_all.html

Tables and drawing

Observations

___________________________________________________________________________

___________________________________________________________________________

___________________________________________________________________________

___________________________________________________________________________

If we keep the angle of incidence fixed and change the medium in which the light refracts, what happens to the angle of refraction?

___________________________________________________________________________

___________________________________________________________________________

Since a law cannot be found, the need to introduce a new concept emerges, let's write to Galadriel.

She suggests going to the human realm to learn how to use a new tool, a sort of 'machine'.

Once you have learnt how to use the machine, you redo the table with the sines of the angles and try again to find a regularity.

**Figure A3.** Quest 6_Part 3.

You change the material and try again, now that you have a better understanding of the relationship between the angles.

Try to write the law.

Rubric for self-assessment: Ability to conduct an observational experiment

| **Is able to describe what is observed without trying to explain, both in words and by means of a picture of the experimental setup.** | No description is mentioned. | A description is incomplete. No labeled sketch is present. Or, observations are adjusted to fit expectations. | A description is complete, but mixed up with explanations or pattern. The sketch is present but is difficult to understand. | Clearly describes what happens in the experiments both verbally and with a sketch. Provides other representations when necessary (tables and graphs). |
|---|---|---|---|---|
| **Is able to identify a pattern in the data** | No attempt is made to search for a pattern | The pattern described is irrelevant or inconsistent with the data | The pattern has minor errors or omissions. Terms proportional are used without clarity - is the proportionality linear, quadratic, etc. but also congruence between angles … | The patterns represents the relevant trend in the data. When possible, the trend is described in words. |

**Figure A4.** Quest 6_Part 4.

**Self-assessment.**                    NAME ________________________

What did I realise when we did the measurements in the lab for Experiment 2? (Think both about the physical content, but also about your thought processes).

_________________________________________________

_________________________________________________

_________________________________________________

What did I realise when we had to write down the observations from the table (experiment 2)? (Think both about the physical content, but also about your thought processes).

_________________________________________________

_________________________________________________

_________________________________________________

What did I realise when we did the simulations? (Think both about the physical content, but also about your thought processes).

_________________________________________________

_________________________________________________

_________________________________________________

What did I realise when we had to write down the observations from the table (experiment 3)? (Think both about the physical content, but also about your thought processes).

_________________________________________________

_________________________________________________

_________________________________________________

What was difficult for me and what did I do to overcome the difficulty?

_________________________________________________

_________________________________________________

_________________________________________________

**Figure A5.** Quest 6_Part 5.

**Quest 7**

*What is the nature of science?*
*What relationship do you see between the*
*discoveries shown at the museum and what we still*
*need to understand about light?*

Names of group participants

_______________________________

Objective: deepen the nature of science through
history; observe an instrument asking questions

A trip offers the opportunity to discover the cultural wonders of our world, our history, our
origins and the importance of encounter, exchange and dialogue.

In the museum, we focus on the following aspects:
- science is a social activity: the ideas put forward by one scientist open up new avenues for
  other scientists, scientists continue what other scientists have thought before them;
- science is a human endeavour (it is linked to the efforts of scientists, their anxieties,
  frustrations, hopes and fears of not being understood).

Take lots of pictures!

Observe an instrument chosen between the burning mirror and the refractometer and ask at
least 30 questions about it.

_______________________________________________
_______________________________________________
_______________________________________________
_______________________________________________
_______________________________________________
_______________________________________________
_______________________________________________
_______________________________________________
_______________________________________________
_______________________________________________
_______________________________________________
_______________________________________________
_______________________________________________
_______________________________________________
_______________________________________________
_______________________________________________
_______________________________________________
_______________________________________________
_______________________________________________
_______________________________________________
_______________________________________________
_______________________________________________

**Figure A6.** Quest 7_Part 1.

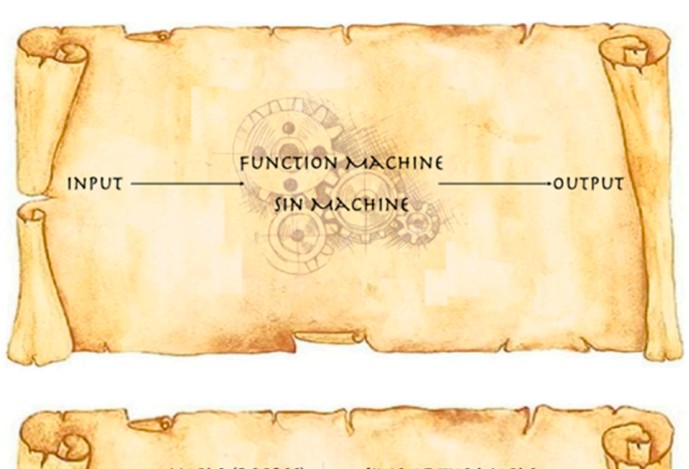

ANGLE (DEGREE) | SINE OF THE ANGLE

| ANGLE (DEGREE) | SINE OF THE ANGLE |
|:---:|:---:|
| 0 | 0.00000000 |
| 5 | 0.08715574 |
| 10 | 0.17364818 |
| 15 | 0.25881905 |
| 20 | 0.34202014 |
| 25 | 0.42261826 |
| 30 | 0.50000000 |
| 35 | 0.57357644 |
| 40 | 0.64278761 |
| 45 | 0.70710678 |
| 50 | 0.76604444 |
| 55 | 0.81915204 |

Name _______________________________

Have you found anything in the museum that you have experienced so far?

_______________________________________________________________________
_______________________________________________________________________
_______________________________________________________________________

What impressed you most about this trip?

_______________________________________________________________________
_______________________________________________________________________
_______________________________________________________________________

How do you think you can use the knowledge found in the parchment?

_______________________________________________________________________
_______________________________________________________________________
_______________________________________________________________________

**Figure A7.** Quest 7_Part 2.

## Quest 8

*How can we apply the law of refraction?*

<u>Objective</u>: solve problems, applying the law of refraction to different situations.

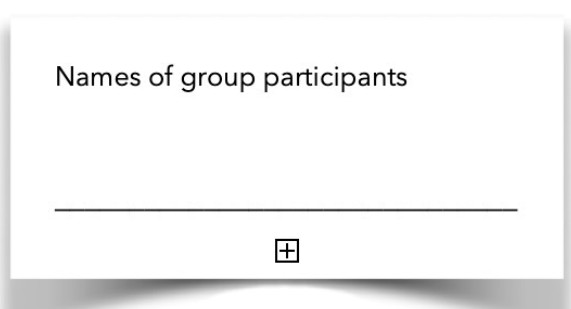

Names of group participants

_________________________________

⊞

### A. Problems to solve

(1) A beam of light passes from a glass with a refractive index of 1.58 into the water with a refractive index of 1.33. The angle of the refracted ray in water is 58.0°. Draw a sketch of the situation showing the interface between the media, the normal line, the incident ray, the reflected ray, the refracted ray, and the angles of these rays relative to the normal line.

(2) A light beam hits the interface between air and an unknown material at an angle of 43° relative to the normal. The reflected ray and the refracted ray make an angle of 108° with respect to each other. What is the index of refraction of the material?

(3) You swim underwater at night and shine a laser pointer so that it hits the water-air interface at an incident angle of 52°. Will a friend see the light above the water? Explain.

(4) The two equations below describe a physical process. Invent a problem for which the equations would provide a solution.
$$1.00 \sin 30° = 1.60 \sin \theta_2$$
$$1.60 \sin \theta_2 = 1.33 \sin \theta_3$$

(5) Imagine that you have a long glass block with a refractive index of 1.56 surrounded by air. Light travelling inside the block hits the top horizontal surface at a 41° angle. What happens next?

### B. Refraction in everyday life

Recognize and describe refraction in real situations (some hints: teaspoon in water, feet in the water, bottom depth, lenses, …)

**Figure A8.** Quest 8.

**Quest 9**

Now it is time to save the Fellowship!

*How to creatively interpret a phenomenon and apply refraction to solve contextualized problems?*

Objective: solve a contextualized open problem by putting into practice the whole knowledge reached about refraction.

Names of group participants

___________________________

What are the characteristics of the material in which the Fellowship is immersed? What do you need and how can you get it?

___________________________________________________________________
___________________________________________________________________
___________________________________________________________________
___________________________________________________________________
___________________________________________________________________
___________________________________________________________________

We have to hit with light (what we see), but where is the switch? How do we aim the light? What happens?

Explain.

If we were to hit the switch (the one we see) with a stick, how do we point the stick? What happens?

**Figure A9.** Quest 9_Part 1.

Name _______________________________

How did you go about solving the problem together?

_______________________________________________

_______________________________________________

_______________________________________________

Which of the above activities was most helpful to you in solving it?

_______________________________________________

_______________________________________________

_______________________________________________

Which skills did you activate to solve the problem?

_______________________________________________

_______________________________________________

_______________________________________________

**Figure A10.** Quest 9_Part 2.

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
