# Peer review of "Adapting the DIST-M Model for Designing Experimental Activities—A Theoretical Discussion from an Interdisciplinary Perspective"

_education, doi:10.3390/educsci14050472_

Round 1
Reviewer 1 Report
Comments and Suggestions for Authors
First of all, thank you for the opportunity to read this manuscript.
This manuscript illustrates a study which aims to answer the following research question: “How and to what extent the experimental features emphasized by design models of inquiry-based learning, can be integrated within the storytelling model proposed by DIST-M?”
Overall, the are of focus of the manuscript is relevant but there are some issues that I think should be addressed, since it lacks some clarifications and justifications, from my perspective.
Introduction
You refer some IBL models (Kolb, 5E, ISLE) (Lines 112-114) without any references regarding these models. From my perspective they should be introduced. In fact, through the manuscript there are missing references when these models are referred.
Models for instructional design
- (Lines 132-133) “From this starting point, to integrate other guides to encompass the physical dimension, from an experimental perspective, we consider Kolb's model.”- Once again there is no reference to what is the Kolb´s model and it is not clear why did the authors choose these model and not another one. This should be clarified.
- Again, and now regarding the DIST-M model, why did the authors choose this one? Are there any other models? Maybe a short overview of some models, and the reason why this one was chosen should be referred.
- Maybe some previous empirical studies should be referred that used these models. Why these ones? What are the evidences that support their choice?
The leaning sequence
- In lines 249-248 the authors refer the learning sequence, but its details are only provided in Table 4. For a better understanding of the reader maybe you should consider moving this table to this part of the manuscript.
- Regarding Table 4, there is something missing in the “Refraction Cycle”
Results
- I am having some trouble with the word “Results”.
As the authors mention in the abstract “The research presents a theoretical discussion grounded in the design of a learning sequence centered around the study of light, taking place in a non-virtual environment and approached from an interdisciplinary standpoint”. As such, this is not an empirical study so the Results section, in my opinion, should be renamed.
Reviewer 2 Report
Comments and Suggestions for Authors
The Manuscript "Adapting DIST-M model for designing experimental activities. A theoretical discussion from an interdisciplinary perspective." is an interesting piece of work at the intersection of storytelling/gamification and interdisciplinarity/modelling/experimentation/inquiry-based learning/family resemblance, which is a refreshing and rich novel approach. It is a very detailed proposal for an extended learning sequence, unifying aspects of mathematics and physics from an 'impartial' perspective, in particular around modelling and experimentation. The work also harmonises the approaches of DIST-M, Kolb's reflective cycle and Udhen' modelling approach by proposing different extension phases. This work is therefore of wider theoretical interest.
The main (though only minor) shortcomings of the work are that the (Italian, school) context only becomes clear somewhat later; and that although it is described as a theoretical discussion, it is does not become clear what the context/purpose of this was (a research degree? A collaboration with teachers and academics?), whether any of it has been tried in reality, whether there are any plans to do so etc.
I think the article warrants publication if the above points are addressed, and below list of suggestions for improvement are considered:
Title: Adapting DIST-M model for designing experimental activities. A theoretical discussion from an interdisciplinary perspective. -> Adapting the DIST-M model ?
p. 1, l. 27: the first word 'interdisciplinary' should probably read 'interdisciplinarity'
l. 36: change , to ;?
p. 2, l. 64: the history-pedagogy-mathematics/physics: the reader really needs a bit more context here to know what this is about
Uhden's modelling cycle: it might be useful to refer to the science modelling work by Hestenes et al as well
l. 83: Liceo Scientifico comes out of nowhere, this needs to be contextualised; there is only mention of the Italian context on the next page
last paragraph: the DIST-M model seems quite specialised; some more general references on storytelling/narrative might be useful here: on narrative theory, Bruner's narrativs vs paradigmatic work, more contemporary science + narrative work (e.g. Scheufele) etc might make the work of wider appeal
p. 3, l. 116: remove , in question?
l. 125: harmonise spelling of Inquiry-based learning
l. 136: the reader would benefit here from a reminder that FRA is family resemblance approach whilst the work of Uhden was on modelling
p. 5, fig 1: the Kolb cycle would probably benefit from directionality, although it is rather obvious
p. 7: the LoTR setting is engaging but it is hardly timely (perhaps it's timeless): could you comment on how portable the 'escape room' puzzles are to different contexts, and what a more timely example might be of what students like these days that could work equally?
p. 8: he/she, his/her: this is somewhat inelegant and unnecessarily binary: they, their seems more straightforward
explore physically refraction -> physically explore refraction or explore refraction physically?
Likewise for mathematically, and similarly in tables T3, bottom of T5 on page 12,
Solve the problem theoretically -> theoretically.
Only one attempt otherwise -> attempt; otherwise
p. 9: T3: search regularities -> search for regularities? Likewise on p. 16 for quest 6
p. 11, l. 350/51: it becomes evident... the needing of adding... please rephrase
p. 19: Fig A4 seems cut off: linear, quadratic, [...?]
p. 20: there is an odd mismatch between 1st and 2nd person: What did I realise vs think about your thought processes. Perhaps this can be resolved by putting the 2nd sentence in brackets throughout?
p. 23: you swim under at night -> you swim underwater at night?
p. 24: Quest 9: where does 'company' suddenly come from? Does it mean 'fellowship'? also inconsistent spelling of it
References:
- harmonise Z. Dagher vs Z. R. Dagher, as well as E.F. Redish vs E. F. Redish etc. My preference would be to have a space between the initials. T. J. Bing, D. T. Brookes ([24, 26])
- the Einstein Infeld book from 2007 seems odd
Altogether an interesting work theoretically, and very imaginative pedagogically. Many thanks.
Reviewer 3 Report
Comments and Suggestions for Authors
Thank you for the opportunity to review this innovative study/paper. It made me excited to think about how physics can make sense using mathematics and vice versa.
I have some comments for your consideration:
Line 102 – I would assume there are similar recommendations in most (if not all) international curriculum documents. Would you consider suggesting this with some examples?
Line 113 – ISLE Acronym needs to be defined.
Lines 115-116 Small edits to research question:
How and to what extent can the experimental features emphasized by design models of
inquiry-based learning, can be integrated within the storytelling model proposed by DIST-M?
Line 177 Is there a reason why Diverging, Assimilating, Converging and Accommodating have not been discussed in the KOLB cycle? What are the significance of these?
Line 364 – “DIST-M helps us to integrate the structural role of mathematics into physics, focusing on reasoning, formalisation, and proof” – make this and other mathematics concepts/learning more evident/explicit in your tables.
Lines 372-373 close bracket )
Lines 459-460 – "Interdisciplinary Interactive Storytelling in Mathematics and Physics" (IIST-MP)” Could you design a visual model of this to assist other colleagues with planning these type of learning experiences?
Line 493 “5E model” – you should explain why this is being considered as an alternative to the Kolb model briefly.
A small discussion on the importance of group work should be included (you may like to read https://buildingthinkingclassrooms.com/).
Did you measure student engagement and student learning? How do you assess individual contribution in group work settings?
Here is an idea for a different topic:
https://www.ted.com/talks/dan_meyer_math_class_needs_a_makeover?language=en
All the very best.
Comments on the Quality of English Language
Minor edits (I have made suggestions above).
Round 2
Reviewer 1 Report
Comments and Suggestions for Authors
Nothing to comment.